# Educational climate of a pathology residency program at a tertiary care hospital

Zafar Ali[1], Hashaam Bin Ghafoor[2], Muhammad Nasir Ayub Khan[3], Muslim Atiq[4], Saira Akhlaq[5]*

1 Division of Pathology, Department of Histopathology, Shifa International Hospital, Islamabad, Pakistan, 2 Department of Anaesthesia, Al-Khor Hospital, Hammad Medical Corporation, Doha, Qatar, 3 Department of Anaesthesia, Shifa International Hospital, Islamabad, Pakistan, 4 Department of Gastroenterology, Shifa International Hospital, Islamabad, Pakistan, 5 Master's in Health Professions Education, Shifa School of Health Professions Education, Shifa Tameer-e-Millat University, Islamabad, Pakistan

* saira.sshpe@stmu.edu.pk

**Data Availability Statement:** Data has been attached as a Supporting Information file.

**Funding:** The author(s) received no specific funding for this work.

## Abstract

Evaluating educational climate (EC) is imperative for ensuring postgraduate trainees' competencies and quality in residency training programs. This study assessed the EC experiences of pathology postgraduate residents (PGRs) during their postgraduate training in pathology residency programs—a cross-sectional study design assigned EC scores in the pathology residency program at a prestigious institution in Islamabad, which were measured using the Dutch Residency Educational Climate Test (D-RECT) questionnaire. Scores from the D-RECT were employed to conduct descriptive statistics and comparison of means across groups to evaluate EC scores by years of training and compared to assess where the differences were located. Among FCPS-II pathology residents, most of whom were females (94.4%), the mean age was 28.11±2.91 years. A mean positive score was observed among all pathology residents ($M \geq 3.6$) for all D-RECT subscales except for the feedback subscale: the average score for feedback was below the average mean score of 3.6 ($M = 3.19$). A significant difference $p = 0.016$ was observed in EC scores across different groups through the Analysis of Variances (ANOVA) test. The most significant difference was between less than two and greater than two groups $p = 0.027$, followed by the difference between equal to two groups and greater than two groups $p = 0.052$. Overall, positive scores for EC in the pathology residency program were observed. Thus, targeted interventions are needed to increase feedback scores and address observed differences in EC scores by years of training.

## Introduction

Educational Climate (EC) is the pivot around which the other four focal areas of medical education (i.e., Curriculum, Environment, Quality, and Change) are considered for discussion [1]. Educational Environment (EE) may be described as an environment that is influenced by the physical environment (i.e., safety, comfort, food), emotional environment (i.e., feedback quality and security), and educational environment (i.e., participation, relevance, and education

**Competing interests:** The authors have declared that no competing interests exist.

planning) [2, 3]. Experiences within an EC are directly or indirectly linked with the valuable outcomes of student achievement, satisfaction, and success [1]. During postgraduate (PG) training programs, EC is an essential marker for quality in PG medical education [4–6], and the successful translation of a curriculum in EE depends upon an adequate EC [7–9]. Enhancing the latter in medical education also depends upon establishing a continuous feedback system, especially among emerging nations and new training institutions [9]. Henceforth, establishing a feedback system in EC of a histopathology residency program is necessary to achieve positive outcomes through an EE that improves learning [7–9], the well-being of PGRs [10–12], professionalism [13], learning satisfaction [14], and competency of health care professionals (HCP) [9]. The EC of PG programs highlights the environment in which PGRs learn about the context and shared identification of approaches, techniques, and practices [15]. In comparison, achievement of excellence through PGRs' training program may be influenced by obtrusive clinical EE that may lead to fatigue, exhaustion, and burnout among PGRs, leading to dropout [16].

Measuring the learning environment (LE) offers a reference point for the didactic institution [17]. The measurement process facilitates answering trainers' and trainees' questions about the learning experience's quality. The pathology training program is considered distinctive because a deficiency is observed in the exposure to several lab sections during house job and MBBS. The pathology residency program is divided into subspecialties such as Chemistry, Histopathology, Immunology, Hematology, and Microbiology; each sub-specialty has its unique EC. To fulfill the increasing demand for pathology training, valuable programs are needed on a large scale as any change or development induces recognition of the available teaching models, boosts PGRs' education, and enhances their experience. The pathology graduates and employers agree that existing pathology residency training remains inadequate in preparing the residents who are competent in clinical chemistry during their professional careers primarily due to less value placed by residents on learning clinical chemistry and the lack of directors who supervise and manage labs of clinical chemistry [18]. The impassiveness of the teaching staff or residents is an essential obstruction in teaching clinical chemistry to the residents [18]. As explained by residents, unsupportive EE and abortive teaching techniques are the main drawbacks [18].

EC using D-RECT has been assessed in various local and international studies [19]. However, these studies involved different disciplines [3, 19–21], and the nature of EC depends upon the nature of the setting, i.e., each unique specialty and sub-specialty [1, 18, 22, 23]. The current study, therefore, assessed the EC of a PG pathology residency program by administering D-RECT and was conducted considering the need for future research in more focused areas by examining the individual disciplines to increase the likelihood of ruling out subject-specific differences [24], mainly when the same climate index can be used even with different stakeholders [7].

## Materials and methods

A cross-sectional study design approved by the ethics committee of Shifa Tameer-e-Millat University was used at Shifa International Hospital (Islamabad, Pakistan) to assess EC scores of pathology residents by years of training. Enrollment in the study required voluntarily informed permission from participants (including all voluntary pathology PGRs), which was acquired through written consent following the Shifa International Hospital Protocol. Data was collected from 06/06/2021-06/30/2021. During this period, the D-RECT instrument (fifty items categorized into eleven subscales) was used to evaluate EC scores among trainees, and demographic data variables like age, sex, specialist area, and years of training were acquired in the data collection process.

**Table 1. Descriptive statistics for educational climate (D-RECT subscales).**

| Subscales | Perspective | | | Mean±SD |
|---|---|---|---|---|
| | Agree | Neutral | Disagree | |
| Supervision | 17 (94.4%) | 1 (5.6%) | 0 (0%) | 4.56±0.457 |
| Coaching and assessment | 18 (100%) | 0 (0%) | 0 (0%) | 4.45±0.429 |
| Feedback | 8 (44.4%) | 10 (55.6%) | 0 (0%) | 3.19±0.458 |
| Teamwork | 18 (100%) | 0 (0%) | 0 (0%) | 4.43±0.427 |
| Peer collaboration | 17 (94.4%) | 1 (5.6%) | 0 (0%) | 4.54±0.606 |
| Professional relations between attending | 18 (100%) | 0 (0%) | 0 (0%) | 4.41±0.465 |
| Work is adapted to residents' competence | 18 (100%) | 0 (0%) | 0 (0%) | 4.10±0.536 |
| Attendings' role | 18 (100%) | 0 (0%) | 0 (0%) | 4.58±0.506 |
| Formal education | 18 (100%) | 0 (0%) | 0 (0%) | 4.19±0.504 |
| Specialty Tutor's role | 18 (100%) | 0 (0%) | 0 (0%) | 4.37±0.497 |
| Patients sign out | 18 (100%) | 0 (0%) | 0 (0%) | 4.22±0.535 |
| Overall D-RECT | 18 (100%) | 0 (0%) | 0 (0%) | 4.39±0.385 |

Data was collected through a paper-format questionnaire stored in locked folders for data safety and security, and paper-format data was converted into electronic files using SPSS 21.0. The mean of the D-RECT ranking and each subscale were determined as follows: the descriptive data analysis included the calculation of Mean ($M$), Standard Deviation ($SD$), frequency ($f$), and percentages (%) and an average score below 3.6 was considered a negative perception in any subscale.

## Results

The response rate was 100% (18). Refer to data in S1 Dataset for complete access to data that was used in the data analysis. Considering the age categories of more than and less than 30 years old, 83.3% (15) of the study participants were ≤30 years, and only 16.7% (3) were >30 years. The average age was 28.11±2.908 years, and PGRs included more females (94.4%) than males (5.6%). PGRs in the first two years of training were more significant in numbers (66.6%) than those PGRs in the last three years (33.4%). The histopathology department had the highest number of PGRs, 55.5% (10), while 33.6% (6), 5.6% (1), and 5.6% (1) were working in the hematology, immunology, and chemical pathology department, respectively.

Overall, positive experiences ($M = >3.6$) were reflected in EC of the pathology residency program as measured through the sub-scales on the D-RECT questionnaire, except Subscale: feedback ($M ≤ 3.6$) in Table 1.

### Comparison of means

**ANOVA.** Data for the overall EC scores was typically distributed as the overall $p > 0.05$. Please refer to the Table in S1 Table for assessing the test of normality that was used to determine the normality of data. An overall summary of the mean scores of EC across three different groups was calculated. There were fewer participants in the created Groups 4 and 5. Therefore, Groups 4 and 5 were added to Group 3 because only one participant was in Groups 4 and 5. There were equal participants in the final groups, i.e., Group 1 (less than 2), Group 2 (equal to 2), and Group 3 (greater than 2). A category "less than 2" refers to PGRs in the 1st year. The "Equal to 2" category refers to PGRs in the second year, and the "greater than 2" category refers to PGRs in the 3rd, 4th, and 5th years. A summary of means across different groups is as follows in Table 2.

**Table 2. Descriptive of variances in EC scores across groups.**

**Descriptive**

**Total LC Scores**

| | N | Mean | Std. Deviation | Std. Error | 95% Confidence Interval for Mean | | Minimum | Maximum |
|---|---|---|---|---|---|---|---|---|
| | | | | | Lower Bound | Upper Bound | | |
| Less than 2 | 6 | 4.59 | .455 | .186 | 4.12 | 5.07 | 4 | 5 |
| Equal to 2 | 6 | 4.53 | .273 | .112 | 4.24 | 4.82 | 4 | 5 |
| Greater than 2 | 6 | 4.05 | .088 | .036 | 3.95 | 4.14 | 4 | 4 |
| Total | 18 | 4.39 | .385 | .091 | 4.20 | 4.58 | 4 | 5 |

One-way ANOVA was conducted to identify if there were any differences between and within groups. A significant difference was observed in the LC scores across different groups, $p = .016$ in Table 3.

**Multiple comparisons.** After observing the ANOVA analysis, multiple comparisons were conducted to identify where the difference was located, highlighted between 1[st] year of pathology and 3[rd] year of residency and above, and it was similarly observed between 2[nd] year of the pathology residency program and 3[rd] year of residency and above in LC scores in Table 4.

## Discussion

EC is a crucial variable that must be monitored to guarantee the caliber of instruction in a healthcare setting that offers an EE for PGR training. Most PGRs in the pathology residency program were females (83.3%) and $\leq$30 years old; generally, numerous study participants (66.6%) were from the first and second years of the pathology residency program.

Considering the subscales on the D-RECT, each may be considered a separate variable or domain for understanding purposes to discuss their unique implications. Additionally,

**Table 3. ANOVA-variations in EC scores by groups.**

**Total LC Scores**

| | Sum of Squares | df | Mean Square | F | Sig. |
|---|---|---|---|---|---|
| Between Groups | 1.073 | 2 | .536 | 5.557 | .016[a] |
| Within Groups | 1.448 | 15 | .097 | | |
| Total | 2.521 | 17 | | | |

[a]The mean difference is significant at the level of 0.05.

**Table 4. Multiple comparisons.**

**Dependent Variable: Total EC scores**

| (I) Years | (J) Years | Mean Difference (I-J) | Std. Error | Sig. | 95% Confidence Interval | |
|---|---|---|---|---|---|---|
| | | | | | Lower Bound | Upper Bound |
| Less than 2 | Equal to 2 | .063 | .179 | .940 | -.42 | .55 |
| Equal to 2 | Greater than 2 | .547[a] | .179 | .027 | .06 | 1.03 |
| | Greater than 2 | .483[b] | .179 | .052 | .00 | .97 |

[a]The mean difference is significant at the 0.05 level.
[b]The mean difference is significant at the 0.05 level.

supervision is one such sub-scale that is an essential component of any training program. The need to supervise pathology residents constantly by hospital teaching staff is always to provide patients with safe, effective, and better-quality care. PGRs score for supervision were high (average score 4.56 +/- 0.457) in a large percentage (94.4%) of study participants, which were comparatively better than the scores (3.75 +/- 1.49) among PGR residents from multiple specialties [20] and during shift change (3.93 +/- 0.36) in a study conducted in the Netherlands that included residents from multiple specialty training programs during shift change [15]. Despite the prevalence of positive perceptions with varying degrees of magnitude in different populations, negative perceptions have also been observed in some studies, specifically about supervision (2.83 +/- 0.83) [4] and in residents of the Saudi emergency department in Riyadh (3.30 +/- 1.17) [23]. Therefore, identifying mechanisms that nurture differences in the magnitude of positive scores and factors/strategies that lead to lower scores for supervision may help address the weaknesses within the educational systems and facilitate building on the strengths of a solid educational system by using it as a model.

Coaching and assessment play a vital role that can never be overlooked. A positive perception of coaching and assessment was prevalent (4.45 +/- 0.429) among study participants (100%), like positive perceptions (3.92 +/- 0.32) observed in medical residents in the Philippines [25]. Despite the prevalence of positive scores in the current study and another study in the literature, there are still some studies that report lower scores (2.60 +/- 0.73, 3.24 +/- 0.35, and 3.08 +/- 1.17) in Saudi psychiatry residents, 45 multi-specialty residents in the Netherlands, and Saudi Emergency residents correspondingly for perceptions about coaching and assessment [4, 15, 23]. Therefore, future studies may identify and compare the factors leading to these scores' differences. Such an approach may help adopt mechanisms associated with higher scores for coaching and assessment.

Feedback is a significant factor that enhances pathology residents' learning process in three ways: it updates learners regarding their deficiency/progress, advises learners regarding observed training needs and resources available, assists them during learning, and encourages learners to engage in adequate learning activities. Despite the well-established understanding of the effectiveness of feedback in learning outcomes if communicated appropriately, feedback scores reportedly had been on the lower end [4, 15, 20]. A negative perception about feedback (3.19 +/- 0.458) in the current study re-affirms the previous findings about the low scores for feedback (2.00 +/- 0.85, 3.24 +/- 0.45 and 2.75 +/- 0.54), respectively [4, 15, 20]. The observed trend reflects the need for further concentration in the focused area of feedback to improve the patient-care capacity of the PGRs. Additionally, significant differences in means of EC scores by years of training signal a need to identify differential pathways through which these differences in means occur. For example, even in the low feedback scores range, lower scores were slightly higher for a single specialty but at multiple sites; perhaps, it could justify the slightly better feedback scores when considering the collective assessment of scores across all specialties of medicine for PG training, even though scores were still lower, i.e., less than 4 (3.75 +/- 1.49) [20].

Similarly, feedback scores remain lower even when the primary purpose is not to measure the feedback scores but to measure feedback scores for assessing their effect. For example, the effect of LE on faculty's teaching performance is assessed by residents' perceptions (3.24 +/-0.45) [15]. When considering feedback scores for each item, it appears that the scores for the generalized feedback to residents are comparatively better (2.80+/-1.18) than the ones from a structured format (2.36+/-1.19) even though the feedback provided through the latter is more regular (2.39+/-1.22), despite the scores remaining consistently lower, i.e., below 4 in any of the three reported ways to provide feedback [23]. Thus, future research studies may utilize these findings and focus more on the needed forms of feedback, like structured rather than

generalized feedback, and identifying mechanisms that nurture the promotion of structured feedback within the educational systems, facilitating the building of a feedback system in educational settings by using this proposed design as a model. This development and implementation of interventions based on conceptual models can be used to improve feedback communication and interpretation to PGRs, requiring the integration of evidence-based methodologies to disseminate structured feedback upon which the learner can reflect.

For PGRs learning, teamwork is considered a significant tool that assists in modifying learners' attitudes and increases their efficiency, yet perceptions about teamwork have been reported differently in literature. In the present study, they are mainly positive (4.43 +/- 0.427), as well as in a study of Saudi emergency residents in Riyadh (4.05 +/- 0.97) [23], while the prevalence of negative ones (2.81 +/- 0.86) has been reported in Saudi psychiatry residents [4]. A large majority of study participants (94.4%) reported high scores for peer collaboration (4.54 +/- 0.606), consistent with the numbers in the literature, i.e., 82% of study participants reported positive perception (4.07 +/- 2.02) in a study of residents from different disciplines in Army Medical Hospital of Rawalpindi, Pakistan [20]. A low score for perceptions regarding peer collaboration was reported in a French population where the score was less than 3.6 (3.54 +/- 0.90) [26], though almost close to 3.6. It may be inferred from the differential prevalence levels for scores indicating teamwork that future interventions may be planned and implemented to enhance teamwork among PGRs.

A positive perception of professional relations among the attendings (4.45 +/- 0.429) was found among all the participants (100%), like high positive perception scores (3.61 +/- 0.67) in a study where students evaluated the effect of learning climate on the faculty's teaching performance [15], and in contrary to the finding in psychiatry residents where negative perceptions (2.71 +/- 0.95) were prevalent [4]. Work is adapted to the residents' competence (4.10 +/- 0.536) was positively perceived among all the pathology residents (100%), despite controversial findings in literature where this variable has been perceived positively (4.06 +/- 0.28) in a study among internal medicine residents in the Philippines [25] and negatively perceived in another study of residents from medical and allied sciences training as well as surgery and allied sciences PGR training of one institution in Pakistan [20]. Perceptions about "Professional relations between the attendings" and "Work adapted to the residents' competence" are two variables that may or may not impact the quality of EC in any healthcare sector. Therefore, consistent efforts should be made to improve the scores for these variables.

Consultants play a vital role in developing positive perceptions about EC amongst residents. A positive perception regarding them (4.58 +/- 0.506) was prevalent among residents (100%); the finding coheres with positive perception (above 3.6) [15] and is contrary to the findings in a study among Saudi psychiatry residents where negative perception prevailed among residents regarding consultant's role (2.71+/-0.86) [4]. An overall trend of positive experiences was observed amongst 100% of pathology residents regarding formal education (4.19±0.504), like a positive perception regarding formal education ($M = >3.6$) amongst residents from multiple PGR training programs in Pakistan [20]. Thus, whether it is the active role of consultants or formal education, consultants may be viewed within the context of formal education; this would help enhance the capacity of consultants in formal education.

The role of teachers can never be underestimated in improving the EC, while successful learning could be attained through the active role of specialty teachers. A positive perception (4.37+/-0.497) for the specialty tutor role was prevalent for 100% of participants, like positive perception (3.83 +/- 0.32) in a validity assessment of the D-RECT tool in a non-westernized context [25]. Despite reporting positive perceptions in different studies, negative perceptions (3.21 +/- 1.02) have also been reported in a study that focused not on a single specialty but multiple specialties [20], resulting in the need for further exploration of specialty teachers' roles.

Views about patient sign-out are essential as a variable because residents' perceptions could improve with an upgraded patient sign-out system. This consideration sparked because of high scores for patients signing out (4.22±0.535) among residents (100%), like the positive scores (3.82 +/- 0.46) during a shift change in a study in the Netherlands that included residents from multiple specialty training programs [15]. Nevertheless, scores in the current study were higher, indicating a positive impact on our patient sign-out system. Despite consistent positive findings in the literature about patient sign-out, negative perception has also been reported (3.50±1.15) when patient sign-out across multiple specialties was done and could reduce the representative size from each specialty [20]. These differential findings provide evidence for further improvement of patient-sign-out systems in specific settings.

Overall, a positive perception (>3.6) about EC Subscales of the D-RECT questionnaire was found in almost all the PGRs working in the four departments (histopathology, chemical pathology, immunology, hematology) of the pathology residency program. A positive perception regarding the overall EC score (4.39+0.385) was observed, which corresponded to the overall positive score for clinical learning (3.85 +/- 0.29) in Filipino medical residents [25]. However, a negative perception regarding EC (3.16±0.92) has also been reported in a study of Moroccan residents where a French translation of the D-RECT scale was used for psychometric analysis of the tool's French version [26]. Despite the positive score for EC, score variations by years of training $p \leq 0.05$ were observed; it is supposed that they reflected changing perceptions of PGRs about the attributes of EC due to the transition of PGRs into senior years. In various years of training, different challenges and milestones may influence perceptions about experiences regarding LC. In senior years (from third to fifth), residents may feel added responsibility with low time availability for each commitment, relatively less supervised task performances, and the increased complexity of competencies that need to be mastered during the residency training [27]. Therefore, the observed differences in EC scores by years of training may reflect that in the earlier years of residency, i.e., first year and second year, residents might be more enthusiastic about their professional futures and invest consistent time commitments for training milestones and adjust corresponding to relatively simple competencies to be mastered.

Additionally, significant differences in means of EC scores by years of training in our study contradict the assessed differences in scores by years of training in an emergency department [23]. However, there are methodological differences in how the differences were calculated. In a study in the emergency department [23], scores were determined individually for each construct on the D-RECT and then compared by years of training. In addition, emergency residency training is only for four years. In our study, we calculated the average mean score for the overall D-RECT and then compared the scores by each year.

Furthermore, in our study, the pathology residency program lasts five years. We condensed these years of training into three groups. We conducted ANOVA to assess if any significant differences could be identified in addition to a post-hoc analysis through multiple comparisons to see where differences were located, as identifying differences in perceptions about EC scores reflects a need to devise mechanisms to address these differences.

A limitation of our study is that it included a smaller sample size. Enrolling any new PGRs for the study was impossible as these were the maximum PGRs during the study, and even though the study participants provided voluntary informed consent to participate, the response rate was 100%. To address the limitation of sampling size, this study may be considered a pilot study and, in the future, may be replicated in the same setting but with a different study design, i.e., longitudinal study design. Our study's strength is that it focuses on the pathology residency program. Considering the diverse needs of EC in each unique setting, it was essential to assess the needs within our EC. To achieve this milestone, a research study was

designed based on methodological rigor to identify differences within our EC, which was necessary to achieve quality assurance in our setting. We categorized the groups not only by the number of participants but also by considering the maturity levels of residents at each level and the complexity of the competencies to be met at different levels.

Furthermore, lower numbers in the senior years that required us to combine the groups may also reflect the program's evolution with the decreasing numbers in the senior years and increasing numbers in the junior years. In the future, changing perceptions of PGRs in pathology residency training programs over time in each group through a longitudinal study design to highlight the dynamics of interactions related to feedback scores may be assessed. Implementing interventions to address the problem of lower feedback scores in educational settings requires a comprehensive plan like training the trainer workshops for supervisors. It would be a short training session in which supervisors practiced giving feedback in a simulated setting that increased the quality of their feedback [28]. Thus, updating the departmental policies to address the identified needs would help increase feedback scores and how the feedback is perceived at the perceivers' end.

## Conclusions

Variations in EC scores were observed by years of training. Despite the observed positive scores for overall EC, variations by years of training reflect the need to identify the factors that lead to these variations. Such an approach would help address the differences in EC scores by implementing targeted interventions. It is expected that feedback scores for pathology trainees can be improved by following the same approach.

## Supporting information

**S1 Dataset. Dataset used in this study.** This is the data file for the data on EC scores in a pathology residency program at a tertiary care hospital.
(CSV)

**S1 Table. Test of normality Kolmogorov-Smirnov.** This is the table for the test of normality that was conducted to confirm if data was normally distributed.
(DOCX)

## Acknowledgments

The research study was initially conducted as a part of the master's in health Professions Education (MHPE) Thesis at Shifa Tameer-e-Millat University.

## Author Contributions

**Conceptualization:** Zafar Ali, Hashaam Bin Ghafoor, Muhammad Nasir Ayub Khan.

**Data curation:** Zafar Ali, Saira Akhlaq.

**Formal analysis:** Zafar Ali, Saira Akhlaq.

**Investigation:** Zafar Ali.

**Methodology:** Zafar Ali.

**Software:** Zafar Ali, Saira Akhlaq.

**Supervision:** Hashaam Bin Ghafoor, Muhammad Nasir Ayub Khan.

**Validation:** Muslim Atiq.

**Visualization:** Zafar Ali.

**Writing – original draft:** Zafar Ali.

**Writing – review & editing:** Saira Akhlaq.

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
