## [Decision Letter · Decision Letter 0]

25 Oct 2023

PONE-D-23-26192Educational Climate of a Pathology Residency Program at a Tertiary Care HospitalPLOS ONE

Dear Dr. Akhlaq,

Thank you for submitting your manuscript to PLOS ONE. After careful consideration, we feel that it has merit but does not fully meet PLOS ONE’s publication criteria as it currently stands. Therefore, we invite you to submit a revised version of the manuscript that addresses the points raised during the review process.

Please submit your revised manuscript by Dec 09 2023 11:59PM. Please include the following items when submitting your revised manuscript:A 'Response to Reviewers' letter that responds to each point raised by the academic editor and reviewer(s). You should upload this letter as a separate file labelled 'Response to Reviewers'.A marked-up copy of your manuscript that highlights changes made to the original version. You should upload this as a separate file labelled 'Revised Manuscript with Track Changes'.An unmarked version of your revised paper without tracked changes. You should upload this as a separate file labelled 'Manuscript'.

We look forward to receiving your revised manuscript.

Kind regards,

Prof. Ritesh G. Menezes, M.B.B.S., M.D., Diplomate N.B.

Academic Editor

PLOS ONE

Journal Requirements:

Reviewers' comments:

Reviewer's Responses to Questions

**Comments to the Author**

1. Is the manuscript technically sound, and do the data support the conclusions?

Reviewer #1: Partly

Reviewer #2: Partly

Reviewer #3: Partly

2. Has the statistical analysis been performed appropriately and rigorously? 

Reviewer #1: I Don't Know

Reviewer #2: I Don't Know

Reviewer #3: Yes

3. Have the authors made all data underlying the findings in their manuscript fully available?

Reviewer #1: No

Reviewer #2: Yes

Reviewer #3: Yes

4. Is the manuscript presented in an intelligible fashion and written in standard English?

Reviewer #1: Yes

Reviewer #2: Yes

Reviewer #3: Yes

5. Review Comments to the Author

Reviewer #1: Overall Review:

This paper presents an interesting study assessing the educational climate (EC) experiences of pathology residents at a hospital in Islamabad, Pakistan using the D-RECT questionnaire. The study finds overall positive perceptions of the EC, except for lower scores on the feedback subscale. There are also significant differences in EC scores by years of training. The paper is well-written and adds to the literature on EC in graduate medical education. I have some suggestions to further strengthen the manuscript:

Minor recommendations:

- In the discussion, expand on possible reasons for the lower feedback scores and differences by training year. Compare to any similar findings in the literature.

- Incomplete phrase (Negative perceptions about supervision have been observed in a study by?)

- Conclusion could be strengthened by providing specific recommendations based on your findings, beyond just suggesting further study of the factors influencing scores.

- Carefully proofread for any typos, grammar issues, or awkward wording.

- Make sure references are in correct journal format.

- Consider adding any limitations of the study design or sample.

Overall, the study makes a valuable contribution to understanding the EC in this residency program. Addressing the suggestions above would further improve the quality and impact of the manuscript.

Based on my review, I did not notice any major scientific or methodological mistakes in the paper. The study design and use of the D-RECT questionnaire seem appropriate. Here are some minor language/writing issues I identified:

Some awkward phrasing, such as: "Despite the prevalence of positive perceptions with varying degrees of magnitude in different populations, prevalence of negative perceptions has also been observed in some studies." This could be smoothed out and clarified.

Typos - "infers" should be "implies", wellbeing of PGRs (extra spaces) etc. Carefully proofread.

Need to be consistent with abbreviations. For example, "post-graduate" is written out sometimes and abbreviated "PG" other times. Pick one format.

Uses words that sound too casual/conversational in academic writing like "bulk" and "infers"

Overall, the writing is pretty clear and understandable. Just needs minor editing for typos, consistency, and using more formal academic style. The paper would benefit from careful proofreading and having someone else review the writing. But I did not see any major scientific, methodological, or statistical issues from my review.

Reviewer #2: The idea is interesting and worth exploring. There are a few things which require further clarification. Firstly, you need to expand on the groups which you compared between (on what basis were these groups created?). Secondly, it would be interesting to see your interpretation for the differences in perceptions and experiences based on your context. I would also be interested to learn of the impact this study has on your program (any policy changes?). Lastly, there is some use of informal language, a few grammatical errors, and some missing information (specifically the name of a study is missing which was alluded to in the discussion).

Reviewer #3: The article is well written. Here are a few comments:

P9, HCP: what is this abbreviation stans for?

P10, inadequate in preparing residents to play their significant role: What is their significant role, could you be more specific?

P 11, results, first two lines: Review numbers (15: young, 15: >30)

P 11, were female: repetitive

P 12, table, Peer collaboration: missing number in neutral

P 15, in a study by However: missing information

Coaching and assessment vs feedback:

In my understanding, coaching and assessment are part of feedback. You cannot coach without giving feedback. I find it difficult that the residents have positive perspective regarding coaching and assessment and at the same time they report the opposite in feedback!

I also have concerns regarding the low number of residents particularly in pathology disciplines other than histopathology and predominantly female participants, which is acknowledged by the authors. I would advise to involve more numbers to be more representative.

6. PLOS authors have the option to publish the peer review history of their article (what does this mean?). If published, this will include your full peer review and any attached files.

Reviewer #1: No

Reviewer #2: No

Reviewer #3: No

---

## [Author Response · Author response to Decision Letter 0]

15 Nov 2023

In response to academic editor: 

This study is not a laboratory protocol. Therefore, I do not need to use those links.

In response to comments regarding editorial requirements:

I have followed the links that describes the PLOS ONE formatting guidelines for the title page, main manuscript, heading fonts, and references and adopted accordingly. If any minor thing is remaining, let me know I will adhere accordingly. 

Reviewer #1: Overall Review:

Minor recommendations:

- In the discussion, expand on possible reasons for the lower feedback scores and differences by training year. Compare to any similar findings in the literature.

- Incomplete phrase (Negative perceptions about supervision have been observed in a study by?)

My answer: I have included possible reasons for variations in feedback scores in lines 174-183.

 I have addressed the issue of incomplete phrases.

- Conclusion could be strengthened by providing specific recommendations based on your findings, beyond just suggesting further study of the factors influencing scores. 

My answer: I have provided recommendations to address low feedback scores through implementation of interventions based on conceptual models. The exact sentences are in the discussion section in line 212-225 and lines 258-268.

- Carefully proofread for any typos, grammar issues, or awkward wording.

My answer- I have tried to address this issue to the best of my knowledge through re-reading multiple times.

- Make sure references are in correct journal format.

My answer- I have followed the PLOS ONE template now.

- Consider adding any limitations of the study design or sample.

My answer- I have included limitations now as a part of discussion.

Here are some minor language/writing issues I identified:

Some awkward phrasing, such as: "Despite the prevalence of positive perceptions with varying degrees of magnitude in different populations, prevalence of negative perceptions has also been observed in some studies." This could be smoothed out and clarified.

My answer- I have re-phrased this sentence. I have almost re-written most of the discussion section by considering the recommendations of one of the reviewers.

Typos - "infers" should be "implies", wellbeing of PGRs (extra spaces) etc. Carefully proofread.

My answer- I have replaced “infers” with “implies”, removed extra spaces before wellbeing of PGRs. 

Need to be consistent with abbreviations. For example, "post-graduate" is written out sometimes and abbreviated "PG" other times. Pick one format.

My answer- I have now written post-graduate (PG) in the beginning and then followed with PG throughout the paper. 

Uses words that sound too casual/conversational in academic writing like "bulk" and "infers"

My answer- Addressed.

Overall, the writing is pretty clear and understandable. Just needs minor editing for typos, consistency, and using more formal academic style. The paper would benefit from careful proofreading and having someone else review the writing. But I did not see any major scientific, methodological, or statistical issues from my review.

Reviewer #2: The idea is interesting and worth exploring. There are a few things which require further clarification. 

Firstly, you need to expand on the groups which you compared between (on what basis were these groups created?). 

My answer- I have now explained in the manuscript that the pathology residency is a 5-year program. Participants in the last three years were comparatively fewer in number. Therefore, I have condensed participants from 5 years of training into three groups for the purpose of study. Group 1 is less than 2 which means years 1, Group 2 is equal to 2 which means year 2, and Group 3 which mean more than 2 is equal to trainees in years 3, 4 and 5. 

Secondly, it would be interesting to see your interpretation for the differences in perceptions and experiences based on your context.

My answer- I have now included the interpretation for the differences in the opening paragraph of the Discussion section. 

 I would also be interested to learn of the impact this study has on your program (any policy changes?). 

My answer- I have now included answer here and a sentence in the discussion section of the manuscript. I believe the differences in the perceptions and experiences relate to the different challenges associated with the milestones to be achieved at each level. I have included that in the manuscript now. Even though policy making has not been updated yet, this evidence can be used to advocate for the needed changes through updated policy. I have made the adjustments accordingly in lines 266-267.

Lastly, there is some use of informal language, a few grammatical errors, and some missing information (specifically the name of a study is missing which was alluded to in the discussion). 

My Answer- I have tried to replace the names of authors and years with descriptions about the studies to which they are referred. 

Reviewer #3: The article is well written. Here are a few comments:

P9, HCP: what is this abbreviation stans for?

My answer- HCP stands for “Health Care Professionals”. I have included it in the manuscript as well. 

P10, inadequate in preparing residents to play their significant role: What is their significant role, could you be more specific?

My answer- I have addressed it in the manuscript. This phrase was awkwardly written. I have re-phrased it. The lines are now as: Pathology graduates and employers agree that existing pathology residency training remains inadequate in preparing the residents who are competent in clinical chemistry during their professional careers primarily due to less value placed by residents on learning clinical chemistry and lack of directors who supervise and manage labs of clinical chemistry [18]. 

P 11, results, first two lines: Review numbers (15: young, 15: >30)

My answer- I have edited this part. It meant a large majority of participants (15) were either less than or equal to 30 years of age, and a smaller number (3) were older than 30 years. 

P 11, were female: repetitive

My answer- Deleted repetitive words. 

P 12, table, Peer collaboration: missing number in neutral

My answer- Addressed. It was 1 participant. 

P 15, in a study by However: missing information

Coaching and assessment vs feedback:

My answer- In my understanding, coaching and assessment are part of feedback. You cannot coach without giving feedback. I find it difficult that the residents have positive perspective regarding coaching and assessment and at the same time they report the opposite in feedback!

I agree that coaching and assessment are part of the feedback process. However, on the D-RECT scale, they are measured as separate constructs. As the coaching and assessment scores are high, and feedback scores are low. I have mostly re-written the entire discussion part as per the recommendations of one of the reviewers to elaborate and focus more on the construct of feedback as this is an important finding that is consistent throughout literature and efforts need to be taken to promote feedback scores rather than just assessing the reasoning behind differences in these scores. 

I also have concerns regarding the low number of residents particularly in pathology disciplines other than histopathology and predominantly female participants, which is acknowledged by the authors. I would advise to involve more numbers to be more representative.

My answer- I agree that the numbers of the study are not generalizable. However, this is the limitation of the study as it was only focused on pathology residency program in one point in time. As the response rate is 100%, there was no option to enroll any more residents at that time. I agree that these are alarming numbers. However, in a lower middle-income country like Pakistan, where availability and disbursement of funds is a major issue; Pathology residency program is at a disadvantaged end. Therefore, it may be considered as a pilot study if due to lack of generalizability issue, there is a publication concern. However, in future longitudinal study designs may be implemented to ensure larger sample sizes in residency programs that have fewer number of residents. Sub-specialties of pathology that have even fewer number of residents would require a separate publication to discuss low enrollment numbers in different sub-specialties of pathology. 

Considering females as major representatives is alarming. However, considering the income prospects after pathology residency program in Pakistan, most of the males do not opt Pathology residency as a first choice. This is also dependent upon the lack of equipment needed to run the clinical chemistry labs. However, a separate comprehensive study would be needed to discuss these aspects.

---

## [Decision Letter · Decision Letter 1]

2 Jan 2024

PONE-D-23-26192R1Educational climate of a pathology residency program at a tertiary care hospitalPLOS ONE

Dear Dr. Akhlaq,

Thank you for submitting your manuscript to PLOS ONE. After careful consideration, we feel that it has merit but does not fully meet PLOS ONE’s publication criteria as it currently stands. Therefore, we invite you to submit a revised version of the manuscript that addresses the points raised during the review process.

Please submit your revised manuscript by Feb 16 2024 11:59PM. If you will need more time than this to complete your revisions, please reply to this message or contact the journal office at plosone@plos.org. Please include the following items when submitting your revised manuscript:A Response to Reviewers' letter that responds to each point raised by the academic editor and reviewer(s). You should upload this letter as a separate file labeled 'Response to Reviewers'.A marked-up copy of your manuscript that highlights changes made to the original version. You should upload this as a separate file labeled 'Revised Manuscript with Track Changes'.An unmarked version of your revised paper without tracked changes. You should upload this as a separate file labeled 'Manuscript'.

We look forward to receiving your revised manuscript.

Kind regards,

Prof. Ritesh G. Menezes, M.B.B.S., M.D., Diplomate N.B.

Academic Editor

PLOS ONE

Reviewers' comments:

Reviewer's Responses to Questions

**Comments to the Author**

1. If the authors have adequately addressed your comments raised in a previous round of review and you feel that this manuscript is now acceptable for publication, you may indicate that here to bypass the “Comments to the Author” section, enter your conflict of interest statement in the “Confidential to Editor” section, and submit your "Accept" recommendation.

Reviewer #1: All comments have been addressed

Reviewer #2: (No Response)

Reviewer #3: All comments have been addressed

2. Is the manuscript technically sound, and do the data support the conclusions?

Reviewer #1: Yes

Reviewer #2: Partly

Reviewer #3: No

3. Has the statistical analysis been performed appropriately and rigorously? 

Reviewer #1: Yes

Reviewer #2: Yes

Reviewer #3: I Don't Know

4. Have the authors made all data underlying the findings in their manuscript fully available?

Reviewer #1: (No Response)

Reviewer #2: Yes

Reviewer #3: Yes

5. Is the manuscript presented in an intelligible fashion and written in standard English?

Reviewer #1: Yes

Reviewer #2: No

Reviewer #3: Yes

6. Review Comments to the Author

Reviewer #1: (No Response)

Reviewer #2: Thank you for adressing my previous comments. There are, however, still a number of grammatical and typographical errors and use of informal language (e.g. it's [line 66], Study was conducted (instead of The study) [line103], writing the word less than and also adding he symbol [line 171]. I believe you need to thoroughly edit the manuscript.

One of the biggest issues I have now with the review is the discussion. The originally submitted discussion was more specific and offered a rich argument. The revised discussion was too general and at times very difficult to follow. I beleive this generalization took away from the strength of the original discussion.

There were also some repatative points made (e.g. the point about poitive perception except in feedback [lines 228-232] and the point about the inability to enroll additional participants).

Reviewer #3: (No Response)

7. PLOS authors have the option to publish the peer review history of their article (what does this mean?). If published, this will include your full peer review and any attached files.

Reviewer #1: No

Reviewer #2: No

Reviewer #3: No

---

## [Author Response · Author response to Decision Letter 1]

2 Feb 2024

In reviewing the comments by the reviewers, I only found the comments by Reviewer # 2. Therefore, responses to reviewer # 2 have been included in this document. 

Additionally, comment by the Academic Editor was to deposit my laboratory protocol in protocols.io. Even though my study is now beyond the protocol stage, I have registered with the protocols.io and entered my abstract for this manuscript without the results section to reserve the doi for this manuscript. Therefore, the reserved doi for this manuscript is: 

DOI: dx.doi.org/10.17504/protocols.io.n2bvj3wk5lk5/v1 (Private link for reviewers: https://www.protocols.io/private/E5424EF9C00211EEAABF0A58A9FEAC02 to be removed before publication.)

6. Review Comments to the Author

Reviewer #1: (No Response)

Reviewer #2: Thank you for addressing my previous comments. There are, however, still a number of grammatical and typographical errors and use of informal language (e.g. it's [line 66], Study was conducted (instead of the study) [line103], writing the word less than and also adding he symbol [line 171]. I believe you need to thoroughly edit the manuscript.

Thank you for your comprehensive feedback. First, I addressed these specific issues in lines 66, 103, and 171. Then, after making adjustments to the content for the other feedback points especially with regards to discussion section, I sent the finalized manuscript to an international professional editor for editing the manuscript in accordance with the standardized English language to prevent any grammatical and typographical errors and use of informal language. 

One of the biggest issues I have now with the review is the discussion. The originally submitted discussion was more specific and offered a rich argument. The revised discussion was too general and at times very difficult to follow. I believe this generalization took away from the strength of the original discussion.

Two of the other reviewers had the issue that the discussion included more content on all domains of D-RECT rather than focusing on the statistically significant findings and their implications. Therefore, considering your comment about the discussion, I have included the same discussion patterns and detailed description of each domain of D-RECT as was in the first draft that you believed added to the beauty of the discussion. In addition, I have retained the content that was requested by the other reviewers with respect to feedback and variations in EC scores by years of training. Therefore, I have tried to the best of my abilities to preserve the feedback of all three reviewers in the final discussion section of the current manuscript version. 

There were also some repetitive points made (e.g. the point about positive perception except in feedback [lines 228-232] and the point about the inability to enroll additional participants).

I have deleted all the repetitive points. Yes, there were specially at two points where the points were repeated about the need for longitudinal study to enroll larger sample size corresponding to the lower sample size in the current study that was based on cross-sectional study design. The point is that the study was completed at one point in time. It was not possible to enroll any more residents at that time as the response rate was 100%, and the new residents would be hired for the residency program in the corresponding year. The researcher had to complete the study in one year as this study was part of the thesis of a Masters’ degree. It does not mean any new additional participants cannot be enrolled. However, for that purpose a longitudinal study would be more feasible. 

 In some qualitative studies with executive leaders the number of participants is much less as compared to the sample in the general population. The same principle applies here that as we only targeted pathology residents in one tertiary care hospital, it was not possible to enroll any more participants. In other words, there were no more eligible participants as our study was not multidisciplinary. It was only aimed at analyzing the educational climate within the pathology residency program context. 

Reviewer #3: (No Response)

---

## [Decision Letter · Decision Letter 2]

26 Apr 2024

Educational climate of a pathology residency program at a tertiary care hospital

PONE-D-23-26192R2

Dear Dr. Akhlaq,

We’re pleased to inform you that your manuscript has been judged scientifically suitable for publication and will be formally accepted for publication once it meets all outstanding technical requirements.

Kind regards,

Prof. Ritesh G. Menezes, M.B.B.S., M.D., Diplomate N.B.

Academic Editor

PLOS ONE

Additional Editor Comments: Kindly address the minor corrections suggested by one of the reviewers at the time of proof corrections or preferably when an opportunity to consider technical corrections is provided by the editorial office.

Reviewers' comments:

Reviewer's Responses to Questions

**Comments to the Author**

1. If the authors have adequately addressed your comments raised in a previous round of review and you feel that this manuscript is now acceptable for publication, you may indicate that here to bypass the “Comments to the Author” section, enter your conflict of interest statement in the “Confidential to Editor” section, and submit your "Accept" recommendation.

Reviewer #4: All comments have been addressed

2. Is the manuscript technically sound, and do the data support the conclusions?

Reviewer #4: Yes

3. Has the statistical analysis been performed appropriately and rigorously? 

Reviewer #4: Yes

4. Have the authors made all data underlying the findings in their manuscript fully available?

Reviewer #4: Yes

5. Is the manuscript presented in an intelligible fashion and written in standard English?

Reviewer #4: Yes

6. Review Comments to the Author

Reviewer #4: The authors have revised the manuscript well. Some suggestions are as under:

1. Page 3, Line 69- Please write [2][3] as [2,3]. Kindly do this throughout the manuscript.

2. Page 3, Line 76- Please expand PGRs as it is used here for the first time.

3. Page 3, Lines 76-79; Reference citation 10 is done after 11-13, 14, 15. Please rectify. Reference citation 10 should follow 9. Accordingly, the reference order and list need to be corrected.

7. PLOS authors have the option to publish the peer review history of their article (what does this mean?). If published, this will include your full peer review and any attached files.

Reviewer #4: No

---

## [Editor Report · Acceptance letter]

13 May 2024

PONE-D-23-26192R2 

PLOS ONE

Dear Dr. Akhlaq, 

I'm pleased to inform you that your manuscript has been deemed suitable for publication in PLOS ONE. Congratulations! Your manuscript is now being handed over to our production team.

Kind regards, 

on behalf of

Professor Ritesh G. Menezes 

Academic Editor

PLOS ONE